# Assessment of self-rated health: The relative importance of physiological, mental, and socioeconomic factors

Dana Hamplová[1]*, Jan Klusáček[1], Tomáš Mráček[2]

**1** Institute of Sociology, The Czech Academy of Sciences, Prague, Czech Republic, **2** Institute of Physiology, The Czech Academy of Sciences, Prague, Czech Republic

* dana.hamplova@soc.cas.cz

**Data Availability Statement:** The data have been deposited to the public data depository (Czech Social Science Data Archive) that is part of the Consortium of European Social Science Data

## Abstract

### Background

The general self-rated health (SRH) question is the most common health measure employed in large population surveys. This study contributes to research on the concurrent validity of SRH using representative data with biomarkers from the Czech Republic, a population not previously used to assess the SRH measure. This work determines the relative contribution of biomedical and social characteristics to an individual's SRH assessment. Studies have already explored the associations between SRH and markers of physical health. However, according to a PubMed systematic literature search, the issue of the *relative* importance of physiological and psychosocial factors that affect individuals' assessments of their SRH has generally been neglected.

### Methodology/Principal findings

Using data from a specialized epidemiological survey of the Czech population (N = 1021), this study adopted ordinary least squares regression to analyze the extent to which variance in SRH is explained by biomedical measures, mental health, health behavior, and socioeconomic characteristics. This analysis showed that SRH variance can be largely attributed to biomedical and psychological measures. Socioeconomic characteristics (i.e. marital status, education, economic activity, and household income) contributed to around 5% of the total variance. After controlling for age, sex, location, and socioeconomic status, biomarkers (i.e. C-reactive protein, blood glucose, triglyceride, low-density lipoprotein, and high-density lipoprotein), number of medical conditions, and current medications explained 11% of the total SRH variance. Mental health indicators contributed to an additional 9% of the variance. Body mass index and health behaviors (i.e. smoking and alcohol consumption) explained less than 2% of the variance.

### Conclusions/Significance

The results suggested that SRH was a valid measure of physiological and mental health in the Czech sample, and the observed differences were likely to have reflected inequalities in bodily and mental functions between social groups.

Archives (CESSDA). It provides open access to social science data for non-commercial use. The data file is accessible at the section "Health Surveys", data ID: CSDA00288 at http://nesstar.soc.cas.cz/webview/.

**Funding:** This work was supported by the Czech Science Foundation (project 22-09220S). Data were acquired through and are deposited in the Czech Social Science Data Archive (ČSDA). The ČSDA research infrastructure project is supported by the Ministry of Education, Youth and Sports within the framework of grant LM2018135. The funders had no role in study design, data collection and analysis, decision to publish, or preparation of the manuscript. The authors thank to all the participants who took part in the study and provided blood samples. They are also grateful to employees of SYNLAB for the blood sampling and analyses of biomarkers and to agency CVVM for questionnaire data collection.

**Competing interests:** The authors declare no competing interests.

## Introduction

The general self-rated health (SRH) question is the most common health measure employed in large population surveys. One reason for its popularity is the assumption that SRH has high validity as a measure of "objective" health [1]. Studies from various countries and social contexts have demonstrated that SRH is a consistent predictor of mortality as the most objective criterion of "true" health [2,3]. Importantly, the predictive power of SRH with respect to mortality persists even after adjusting for more objective indicators of health, such as biomarkers [4,5]. However, despite the general acceptance of SRH, there is evidence that response styles and validity of SRH might vary across countries [6,7] and that this indicator might be problematic when used as a measure of "true" health in some cases [6–9]. Thus, it is imperative to explore the validity and meaning of SRH in different social contexts and countries and not to assume its validity based on samples from elsewhere.

Moreover, although SRH is widely used, the discussion on its meaning continues. On the one hand, SRH is consistently associated with many indicators of physical health, including cardiovascular diseases, glycemic markers, markers of the autonomic nervous system, hemoglobin, white cell counts, blood pressure, cholesterol levels, BMI, and inflammatory markers [10–19]. For this reason, SRH is viewed as a reliable and valid measure of illness and objective medical burden. On the other hand, growing empirical evidence shows that individuals' assessments of their own health are contingent on their social experiences [20,21]. Thus, studies have demonstrated that a respondent's perception of health and how they respond to SRH questions might be affected by their health expectations, sex, culture, personality, education, social norms, believing that their work is meaningful, self-concept of being a healthy or unhealthy person, and other factors [6,22–25].

This paper offers two contributions to the existing literature. First, it raises a fundamental question of the validity of SRH using a wide range of indicators. Although studies have tested the concurrent validity of SRH based on physiological or psychosocial correlates, there is a dearth of research on the *relative* importance of these domains. In general, studies have tended to focus on either the association between SRH and "objective" markers of physical health [10–16] or the association between SRH and various socio-demographic characteristics [14,20,26]. However, according to a systematic PubMed literature search, the relative importance of the physiological and psychosocial factors that affect individuals' assessments of their SRH has not yet been investigated. Therefore, the following analyses were conducted to determine how much variance in SRH can be explained by biomedical, psychological, and social indicators. This is an important issue, as SRH has been widely used in previous studies as a measure of the social determinants of health and as an indicator for measuring social inequalities in health [27–31]. The common assumption of these studies was that differences in SRH reflect inequalities in "true" health [32]. If this analysis showed that SRH is predicted more by some social characteristics rather than by direct measures of health, such a result would warrant caution in dealing with SRH.

Second, the study used data from the Czech Republic. To our knowledge, no study has tested the concurrent validity of SRH, i.e. the extent to which this indicator correlates with established measures of health, using biomedical data in this country. We believe that it is important to analyze the validity of SRH in different contexts as response styles and predictive power of SRH significantly vary across countries. In the Czech Republic, SRH has been studied in various national and comparative studies, including studies of the general population [33–37], immigrants [38], and school-aged children [39] but these studies did not address the issue of the validity of this indicator. The validity of SRH was tested by Baćak and Ólafsdóttir [40] using data from the 2014 European Social Survey, which included data from the Czech Republic. However, the study observed relied exclusively on self-reports of health problems and did

not estimate the relative contribution of biomedical, mental, and social correlates. In the current study, we report associations between SRH and various measures of health, including biomarkers, and we focus on the proportion of SRH variance explained by these measures. Thus, we adopt Borsboom et al.'s [41] concept of validity maintaining that an indicator is a valid measure of the outcome if the indicator produces variations in the outcome.

## Materials and methods

### Study population and design

The study population was defined based on the QUALITAS—Wellbeing in health and disease survey. A total of 1056 individuals, aged 18 years or older, residing in Prague (capital, 1.4 million inhabitants) and České Budějovice and surroundings (100,000 inhabitants) in Southern Bohemia, participated in the study. They were selected for the face-to-face interviews via quota sampling (i.e. sex, age, education, place of residence, and community size) based on the 2011 Czech population and housing census. The study followed the principles of the Declaration of Helsinki and was approved by the Ethics Committee of the Institute for Clinical and Experimental Medicine and Thomayer Hospital in Prague (study number G-16–05–02). Written informed consent was obtained from each participant who provided blood samples prior to enrolment in the study after an explanation of the study procedures.

The participants were asked to provide a fasting blood sample and to participate in the survey related to their health and socioeconomic status. The questionnaires were administered via face-to-face interviews. Because the participants were selected by quota sampling, there were no missing values for sex, age, education, place of residence, and community size. As for other covariates (i.e. biomarkers, reported health problems, economic activity, sleep quality, alcohol consumption, and smoking), the proportion of missing values was small ($< 1\%$). In total, 35 respondents ($< 3.5\%$ of the sample) were dropped from the analysis because of missing information for at least one of these variables.

Personal income, the only variable with a large number of missing values (21%), was dealt with as follows. Initially, the model was only estimated for respondents who had answered the question. However, to acknowledge that a subsample with non-missing values differed from the full sample—the refusal was more common among men and the economically active population—two other strategies were employed. To deal with a large amount of missing data, the multiple imputation method was employed. This method, an iterative form of stochastic imputation, uses the distribution of observed data to estimate multiple values for missing information. Multiple plausible values are produced to reflect the uncertainty of the true value [42]. However, as this study primarily addressed how much SRH variance is explained by various sets of indicators, the standard method of applying multiple imputations cannot be used due to limitations in calculating the share of explained variance ($R^2$) in imputed datasets. Thus, we did not use the full imputation model. Instead, we used the multiple imputation method to produce 25 plausible values for the missing responses for personal income and calculated the mean of these plausible values, which was subsequently entered into an ordinary least squares (OLS) regression. The disadvantage of this approach is that it fails to account for uncertainty due to the missing information. Thus, for the final step, we used all 25 imputed values to estimate 25 regression models to produce 25 "plausible" values for the explained variance ($R^2$). The distribution of this new variable was then reported (see S1 Appendix).

### Sample characteristics

The characteristics of the analytical sample are shown in Table 1. The ages ranged from 18 to 94, with a mean age of 44.6 (SD 16.0). Compared with the 2016 population statistics [43],

**Table 1. Sample characteristics.**

| Age | 18–29 | 22.4 |
|---|---|---|
| | 30–44 | 29.4 |
| | 45–59 | 26.1 |
| | 60+ | 22.2 |
| | Mean | 44.6 |
| Sex | Male | 42.3 |
| | Female | 57.7 |
| Education | Primary | 8.0 |
| | Occupational | 23.5 |
| | HS | 40.9 |
| | Tertiary | 27.6 |
| Marital status | Single | 37.4 |
| | Married | 39.3 |
| | Cohabiting | 23.3 |
| Economic activity | Not in labor force | 36.2 |
| | In labor force | 63.8 |
| Personal income | No income | 5.0 |
| | up to 9 999 CZK | 11.3 |
| | 10 000–19 999 CZK | 36.4 |
| | 20 000–29 999 CZK | 20.4 |
| | 30 000–49 999 CZK | 5.5 |
| | 50 000+ CZK | 1.2 |
| | No answer | 20.1 |

Source: QUALITAS 2016/2017 survey (N = 1021).

where the mean age of the adult population in Prague and South Bohemia was 48.5, our sample was slightly younger. This might be partly due to not targeting an institutionalized population, only those living in private dwellings. In a supplementary analysis (not included here), we re-ran the models with an upper age limit of 80, but there was no difference in the results compared with using the age-restricted sample. Compared to the population statistics [43], women were overrepresented in our sample (57.7% in the QUALITAS sample and 51.8% in the population statistics). All the models controlled for age, sex, and location (i.e. Prague vs. South Bohemia).

As for the other sample characteristics, 8.0% of the respondents did not finish any type of high school, while 27.6% held a tertiary level degree. Compared to the 2011 Census, our sample slightly underrepresented lower educational groups and overrepresented those with general secondary and tertiary educations. Furthermore, 63.8% of the respondents were economically active, whereas 36.2% were out of the labor force. This category incorporated mainly retirees and students, over two-thirds of the non-active population, but also included women on parental leave, housewives, the unemployed, and individuals on disability pensions. Of the respondents, 37.4% were not living with a partner, 39.3% were married and living with their spouse, and 23.3% were living with a partner without being married (see Table 1).

### Dependent variable

Self-rated health was assessed using a single item: "How is your health in general? Would you say your health is . . ." The response categories were 1) *very good*, 2) *good*, 3) *fair*, 4) *bad*, and 5)

*very bad*. As only two people reported very bad health, the last two categories were merged. In the regression models, the scale was reversed, so higher values indicated better health.

## Independent variables

The independent variables were divided into the following categories: self-reported measures of physical health, self-reported measures of mental health, health behaviors, indicators of socioeconomic status, and biomarkers.

To evaluate the respondents' health status, a list of seven common health conditions was assembled, and the study participants were asked to state whether they had even been diagnosed with the conditions (*yes* or *no*). The list comprised the following items: high level of cholesterol; cardiovascular problems (including heart attack or coronary thrombosis); stroke or any kind of cerebrovascular accident; diabetes or high blood sugar; Parkinson's disease; liver conditions or liver cirrhosis; and cancer or malignant tumor, including leukemia and lymphoma (except for minor skin tumors). Using these items, a summary index indicating the number of diagnoses was produced (min = 0, max = 3; mean = 0.47, and SD = 0.76). In the supplementary analysis, we tested the possibility that the SRH was affected not only by the number of health conditions but also by the specific combination of conditions. However, this hypothesis was not confirmed, and the number of conditions was clearly shown to be the best predictor of SRH.

Furthermore, the respondents were asked to report all the medications they were currently taking. The reported medications were coded into 50 drug classes. This information was used to calculate the number of drug classes that the respondent was being treated with.

In addition, the respondents provided information on their height and weight, from which their body mass index (BMI) was derived (min = 16.0, max = 50.7, mean = 26.1, and SD = 4.9).

Self-reported mental health was assessed with four items adapted from the Centre of Epidemiological Studies Depression (CES-D) scale, which is commonly used to measure depressive symptoms in large population surveys [44]. The respondents were asked: "How much of the time during the past week . . . you felt depressed; you felt that everything you did was an effort; you felt sad; you felt that you could not get going?" The response categories were 1) *none or almost none of the time*, 2) *some of the time*, 3) *most of the time*, and (4) *all or almost all of the time*. Cronbach's alpha confirmed that these items had high internal consistency (alpha = 0.80). The original CES-D scale contains an item on sleep quality. While this indicator was not used in the QUALITAS study, the dataset included the question, "How do you rate the quality of your sleep?" The response categories were 1) *very good*, 2) *rather good*, 3) *rather bad*, and 4) *very bad* (mean = 2.00; SD = 0.773). Even though this variable differed from the original CES-D sleep quality item, we tested the possibility of including it among the mental health measures. This decision was motivated not only by the full CES-D scale containing a sleep item but also by existing research demonstrating that sleep disturbances and mental health are closely related [45–47]. To validate the scale, we ran a measurement model using SEM confirmatory factor analysis, which confirmed very high internal consistency (RMSEA 0.038; CFI 0.996; TLI 0.989; see Table A1 in S1 Appendix). Thus, to create a single indicator of mental health, we used the predicted values from this model (min = −0.713; max = 1.7000; mean = 0; SD = 0.46). The higher the value, the more frequent the participants' depressive symptoms.

We also included some behavioral indicators likely to be linked to SRH, including, in particular, smoking and alcohol consumption. In the sample, 25.3% of the respondents reported current smoking. Former smokers were coded as non-smokers. Alcohol consumption was measured by the number of events when the respondent had felt strongly under the influence of alcohol in the last 6 months. Given that the variable was highly skewed, it was recorded as a

categorical variable with four levels: *never* (57.5%), *once* (15.8%), *two to five times* (18.1%), and *six times or more often* (8.6%). In addition, BMI was incorporated into the analysis (mean = 26.0; SD = 4.9).

Socioeconomic status was measured using four indicators. The highest level of education was coded using four categories: primary (comparative category), occupational secondary school, general secondary education, and tertiary/university education. These categories reflected the main divisions of the Czech educational system. Marital status was coded using three categories: single as a comparative category (i.e. those currently not living with a partner irrespective of their formal marital status), married and living with a spouse, and unmarried cohabitation (i.e. living with an unmarried partner). Employment status dichotomized respondents into two categories: working (coded as 1) and non-working (coded as 0). Given the very low level of unemployment in the Czech Republic, particularly in the locations where the data were collected ($< 2\%$), it was not possible to further distinguish between various types of inactivity.

The respondents' monthly net income was measured using 14 categories, which were treated as a linear expression of the underlying income distribution.

## Biomarkers

After completing the questionnaire, the participants were invited to the local branch of a commercial laboratory (Synlab) to provide a fasting blood sample. C-reactive protein (CRP), blood glucose, and blood lipids (triglycerides [TG], low-density lipoprotein [LDL], and high-density lipoprotein [HDL]), were determined using routine laboratory analyses.

Two analytical approaches were adopted for the biomarkers. First, we used linear measures for all the biomarkers. The means and standard deviations are reported in Table 2. Graphs showing the distribution of the biomarkers are reported in Fig A1 in S1 Appendix. Second, we produced a set of binary variables that distinguished values under and above the reference level for each indicator.

C-reactive protein (CRP) is an indicator of inflammation and cardiovascular disease (mg/L). While some laboratories are limited by their lower levels of detection [14], this was not our case as we were able to detect CRP levels <1 mg/L. CRP levels >5 mg/L are considered to be high risk and CRP levels >10 mg/L suggest recent or ongoing infection [48]. Seven observations of unusually high CRP values (>34) were dropped from the analysis to avoid possible bias in the regression analysis. To indicate the C-reactive protein risk status, we distinguished between CRP $\geq$ 5 (coded as 1) and CRP $<$ 5 (coded as 0).

Fasting blood glucose is a marker of diabetes or prediabetes (mmol/L). Fasting blood glucose levels >5.6 mmol/L indicate prediabetes, and glucose levels over 7 nmol/L suggest

**Table 2. Distribution of biomarkers in the analytical sample as a total and by sex.**

| | All | | Men | | Women | |
|---|---|---|---|---|---|---|
| | Mean | SD | Mean | SD | Mean | SD |
| Glukose (mmol/L) | 5.25 | 1.16 | 5.39 | 1.23 | 5.13 | 1.06 |
| CRP (mg/L) | 3.05 | 3.96 | 2.59 | 3.64 | 3.39 | 4.17 |
| TG (mmol/L) | 1.43 | 1.07 | 1.66 | 1.19 | 1.27 | 0.95 |
| HDL (mmol/L) | 1.46 | 0.36 | 1.30 | 0.29 | 1.59 | 0.36 |
| LDL (mmol/L) | 3.37 | 0.86 | 3.40 | 0.85 | 3.35 | 0.86 |
| LDL/HDL ratio | 2.43 | 0.84 | 2.73 | 0.86 | 2.21 | 0.74 |

Source: QUALITAS 2016/2017 survey (N = 1021).

diabetes [49]. To indicate the glucose status risk, we distinguished between those with a fasting blood glucose $\geq$ 5.6 (coded as 1) and those with lower levels of glucose (coded as 0).

Triglycerides are a type of fat found in the blood, and as converted calories are stored in the fat cells, TG are predictive of cardiovascular disease [50]. Concentrations >2 mmol/L suggest an increased risk of cardiovascular disease, and concentrations >10 mmol/L indicate an increased risk of acute pancreatitis and, possibly, cardiovascular disease. We dropped one observation from the analysis because it was implausibly high (16 mmol/L). The binary indicator for the triglycerides status contrasted TG $\geq$ 2 mmol/L with lower levels of triglycerides.

High-density lipoprotein is involved in the transport of cholesterol from peripheral tissues to the liver. Also, HDL particles have anti-oxidant, anti-inflammatory, anti-thrombotic, and anti-apoptotic properties [51]. Reduced HDL cholesterol concentration has been correlated with numerous risk factors, including components of metabolic syndrome [52]. The reference value for HDL cholesterol is 1 mmol/L. We distinguished between those with HDL < 1 mmol/L (coded as 1) and those with HDL $\geq$ 1 mmol/L (coded as 0).

Low-density lipoprotein carries the majority of the cholesterol in the circulation [51]. It is considered to be the 'bad cholesterol' and high LDL cholesterol levels are associated with a higher risk of cardiovascular disease. The level considered to be 'good' in healthy people is below 3.4 mmol/L. To indicate the LDL risk status, we distinguished those with LDL $\geq$ 3.4 mmol/L (coded as 1) and those with HDL < 3.4 mmol/L (coded as 0).

The LDL/HDL ratio as a risk indicator has a greater predictive value than the use of isolated parameters (i.e. LDL, HDL, and total cholesterol) [52]. Individuals with a high LDL/HDL ratio have greater cardiovascular risk due to the imbalance between the cholesterol carried by LDL, which carries most of the cholesterol in the circulation, and protective HDL. The LDL/HDL ratio should not be >3.5 for men and >3 for women. Thus, we distinguished between higher and lower LDL/HDL ratios for men and women, respectively.

Using these given reference limits, a composite measure indicating several biomarker levels outside the "normal" range was produced. We were aware that the observed biomarker levels might be affected by the medication the respondents were taking. Thus, we produced an adjusted set of biomarkers that combined information from the blood samples with medications. For example, if a respondent was on anti-diabetes medication, the glucose level was coded as over the limit, even if it was within the healthy range. These adjusted biomarkers were used to produce an adjusted summary index that indicated the number of biomarkers outside the normal healthy range.

## Statistical analysis

All the analyses were carried out using STATA, Version 16 (StataCorp). To identify the extent to which health status, mental health, and social factors contribute to variations in SRH, a set of OLS regressions was run to estimate the proportion of variance in SRH explained by each factor. The proportion of explained variance was expressed by $R^2$, which was calculated as 1 minus the proportion of unexplained variance (i.e. the variability of the dependent variable that is not predicted by the model divided by the total variability of the dependent variable).

Our work is inspired by Hiyoshi et al.'s [53] approach to estimating the joint contribution of several variables [also see 54,55]. Similarly to these authors, we analyze contributory factors that are measured by a set of variables, not a single item. The block of variables representing the given contributory factor is entered into the model in a stepwise manner. However, unlike these studies, we are not interested in the question of how various contributory factors attenuate an effect of another explanatory variable. In this study, we focus on the proportion of explained variance of SRH.

Assuming that our SRH measure represented an underlying continuous concept of subjective health and given that dichotomizing the variable would lead to a significant loss of information, we treated SRH as a continuous variable. Moreover, given our aim of estimating the proportion of variance in SRH explained by various factors, we could not use logistic regressions for the binary dependent variable. The pseudo-$R^2$ derived from these models could not be interpreted as a proportion of explained variance because these models were produced using the maximum likelihood method and not calculated to minimize variance. A number of studies on the validity of SRH from other contexts have also used OLS regression [14,40,56–58]. The diagnostics for the reported models are available in Figs A2-A4 in S1 Appendix.

## Results

### Bivariate analysis results

Fig 1 displays the $R^2$ for each variable and reports the proportion of explained variance from a set of independent models that controlled for age, sex, and location, using binary biomarker measures, both raw and adjusted for medication status. Fig 1 demonstrates that mental health was by far the most important predictor of SRH. By not considering other predictors, mental health explained around 14% of SRH variance. This indicator was followed by the self-reported number of medical conditions (confirmed medical diagnosis) and medications (number of drug classes). Both indicators explained around 7% of the variance. In contrast, socioeconomic characteristics (except for education) only had a weak effect on SRH (around 1% of each explained variance).

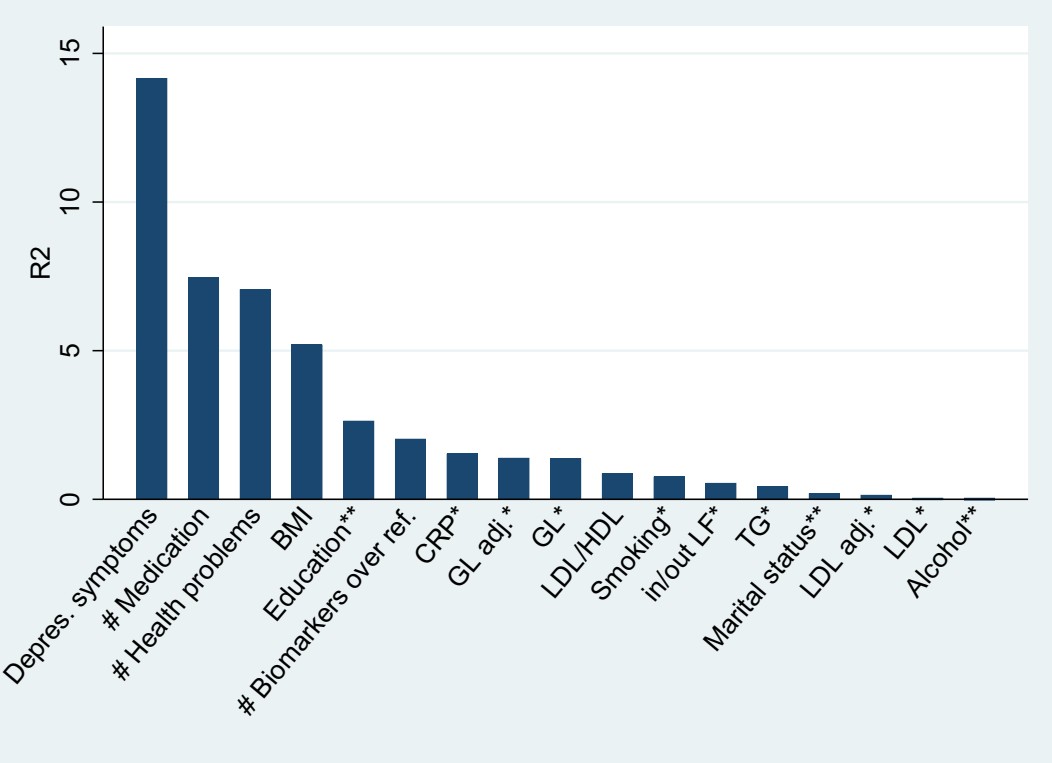

**Fig 1. The proportion of self-rated health (SRH) variance ($R^2$) explained by each variable after controlling for age, sex, and location (N = 1021).** Note: * binary indicator, ** categorical indicator, Adj. values = values adjusted for medication.

**Table 3. ANOVA test for biomarker differences by SRH category, predicted values, and contrasts between categories.**

| | Anova test | | | | | | |
|---|---|---|---|---|---|---|---|
| | SS | F | Prob>F | R2 | | | |
| CRP—C-reactive protein | 553.93 | 12.28 | 0.000 | 0.053 | | | |
| Glucose | 22.50 | 7.50 | 0.000 | 0.108 | | | |
| TG—triglycerides | 10.99 | 3.66 | 0.018 | 0.058 | | | |
| LDL | 10.07 | 3.36 | 0.002 | 0.100 | | | |
| HDL | 3.30 | 1.10 | 0.000 | 0.203 | | | |
| LDL/HDL | 18.83 | 6.28 | 0.000 | 0.157 | | | |
| | Predicted values | | | | Contrasts (Prob > F) | | |
| | 1 Bad | 2 Fair | 3 Good | 4 Very good | 1:2 | 2:3 | 3:4 |
| CRP—C-reactive protein | 5.89 | 3.66 | 2.76 | 2.37 | *0.000* | *0.003* | 0.225 |
| Glucose | 5.55 | 5.46 | 5.14 | 5.13 | 0.621 | *0.000* | 0.945 |
| TG—triglycerides | 1.72 | 1.53 | 1.43 | 1.26 | 0.275 | 0.213 | *0.057* |
| LDL | 3.34 | 3.35 | 3.47 | 3.21 | 0.956 | *0.068* | *0.000* |
| HDL | 1.39 | 1.40 | 1.46 | 1.57 | 0.892 | *0.021* | *0.000* |
| LDL/HDL | 2.57 | 2.50 | 2.51 | 2.17 | 0.583 | 0.918 | *0.000* |

Note: Df = 3. All models were controlled for age, sex, and location. Multivariate analysis results.

Source: QUALITAS 2016/2017 survey (N = 1021).

The difference in biomarkers by SRH levels tested using analysis of variance (ANOVA) is presented in Table 3. All models control for age, sex, and location. This table demonstrates that, with exception of triglycerides, respondents reporting different levels of SRH significantly differed in their biomarker levels. Those reporting bad or fair health had higher levels of C-reactive protein and glucose and lower levels of HDL. In contrast, those in very good health had significantly lower LDL/HDL ratios.

## Multivariate analysis results

While Fig 1 is useful for descriptive purposes, it does not provide an answer to the question of how much SRH variance is explained by bodily conditions, mental health, health behavior, and socioeconomic factors. For example, age and the number of health problems were correlated (Spearman's = 0.36; $P < 0.0001$), but their contribution could not be interpreted in an additive manner. Therefore, we ran multivariate models. First, we focused on the block of variables independently: socioeconomic characteristics, biomarkers, medication, self-reported health measures, mental health indicators, and health behavior. The results from these regressions and robustness checks are reported in (Tables A2–A4 in S1 Appendix). Second, we estimated models that took all these factors simultaneously (see Table 4).

All models used SRH as the dependent variable, and higher values indicated better health. All tables show standardized (beta) and unstandardized coefficients, standard errors, significance tests (t-test: * $p < 0.05$; ** $p < 0.01$), Bayesian information criterion (BIC), the proportion of explained variance ($R^2$), and $R^2$ adjusted for degrees of freedom.

Table 4 integrated all blocks of variables (socioeconomic characteristics, biomarkers, medication, self-reported health measures, mental health indicators, and health behavior) in a stepwise manner. Model 1, serving as a baseline model, controlled for age, sex, and location. Age was the only variable that was significantly linked to SRH, and an age difference of 10 years was associated with a 0.2 shift in SRH. Sex and location (Prague vs. South Bohemia) were not

**Table 4. Results of ordinary least squares (OLS) regression models with dependent variable SRH, displaying regression coefficients, standardized coefficients (beta), standard errors (in parentheses), and significance level.**

| | M1 | | M2 | | M3 | | M4 | | M5 | | M6 | |
|---|---|---|---|---|---|---|---|---|---|---|---|---|
| | Coef. | Beta | Coef. | Beta | Coef. | Beta | Coef. | Beta | Coef. | Beta | Coef. | Beta |
| Age | -0.020** | -0.395** | -0.021** | -0.414** | -0.019** | -0.380** | -0.012** | -0.246** | -0.014** | -0.274** | -0.013** | -0.253** |
| | (-0.001) | | (-0.002) | | (-0.002) | | (-0.002) | | (-0.002) | | (-0.002) | |
| Male | 0.019 | 0.011 | 0.000 | 0.000 | 0.065 | 0.040 | 0.041 | 0.025 | -0.028 | -0.017 | -0.023 | -0.014 |
| | (-0.047) | | (-0.048) | | (-0.052) | | (-0.050) | | (-0.047) | | (-0.048) | |
| Location | 0.031 | 0.019 | -0.047 | -0.029 | -0.028 | -0.017 | 0.014 | 0.009 | 0.031 | 0.019 | 0.026 | 0.016 |
| | (-0.047) | | (-0.047) | | (-0.047) | | (-0.045) | | (-0.042) | | (-0.042) | |
| Married (ref. single) | | | 0.033 | 0.02 | 0.062 | 0.038 | 0.051 | 0.031 | 0.025 | 0.015 | 0.009 | 0.005 |
| | | | (-0.055) | | (-0.054) | | (-0.051) | | (-0.048) | | (-0.048) | |
| Cohabiting (ref. single) | | | -0.027 | -0.014 | -0.012 | -0.006 | -0.012 | -0.006 | -0.041 | -0.021 | -0.037 | -0.019 |
| | | | (-0.062) | | (-0.061) | | (-0.058) | | (-0.054) | | (-0.054) | |
| Occupational secondary | | | -0.072 | -0.038 | -0.084 | -0.044 | -0.063 | -0.033 | -0.069 | -0.036 | -0.074 | -0.039 |
| (ref. primary education) | | | (-0.098) | | (-0.097) | | (-0.092) | | (-0.086) | | (-0.086) | |
| General secondary | | | 0.147 | *0.09* | 0.116 | 0.070 | 0.140 | 0.085 | 0.120 | 0.073 | 0.107 | 0.065 |
| (ref. primary education) | | | (-0.094) | | (-0.092) | | (-0.088) | | (-0.082) | | (-0.082) | |
| Tertiary | | | 0.212* | 0.118* | 0.172 | 0.095 | 0.211* | 0.117* | 0.168 | 0.093 | 0.146 | 0.081 |
| (ref. primary education) | | | (-0.099) | | (-0.097) | | (-0.093) | | (-0.086) | | (-0.086) | |
| Economically active | | | -0.037 | -0.022 | -0.028 | -0.017 | -0.082 | -0.049 | -0.064 | -0.038 | -0.034 | -0.020 |
| | | | (-0.060) | | (-0.059) | | (-0.057) | | (-0.053) | | (-0.053) | |
| Income (imputed) | | | 0.042** | 0.161** | 0.037** | 0.143** | 0.030** | 0.115** | 0.023** | 0.088** | 0.022** | 0.086** |
| | | | (-0.010) | | (-0.009) | | (-0.009) | | (-0.008) | | (-0.008) | |
| CRP—C-reactive protein | | | | | -0.022** | -0.107** | -0.014* | -0.069* | -0.015** | -0.075** | -0.010 | -0.048 |
| | | | | | (-0.006) | | (-0.006) | | (-0.005) | | (-0.005) | |
| Glucose | | | | | -0.057** | -0.080** | -0.020 | -0.028 | -0.006 | -0.008 | 0.000 | -0.001 |
| | | | | | (-0.021) | | (-0.021) | | (-0.019) | | (-0.019) | |
| TG—triglycerides | | | | | -0.007 | -0.009 | 0.007 | 0.010 | -0.001 | -0.001 | 0.013 | 0.018 |
| | | | | | (-0.025) | | (-0.024) | | (-0.022) | | (-0.023) | |
| LDL–low density | | | | | 0.073 | 0.077 | 0.036 | 0.038 | 0.019 | 0.020 | -0.001 | -0.001 |
| lipoprotein | | | | | (-0.040) | | (-0.038) | | (-0.036) | | (-0.036) | |
| LDL/HDL RATIO | | | | | -0.135** | -0.140** | -0.098* | -0.102* | -0.073 | -0.076 | -0.032 | -0.034 |
| ——— | | | | | (-0.046) | | (-0.044) | | (-0.041) | | (-0.042) | |
| Medication # | | | | | | | -0.128** | -0.202** | -0.113** | -0.178** | -0.104** | -0.164** |
| | | | | | | | (-0.020) | | (-0.019) | | (-0.019) | |
| Diagnoses # | | | | | | | -0.195** | -0.184** | -0.146** | -0.137** | -0.144** | -0.135** |
| | | | | | | | (-0.033) | | (-0.031) | | (-0.031) | |
| Depressive | | | | | | | | | -0.550** | -0.312** | -0.534** | -0.303** |
| | | | | | | | | | (-0.045) | | (-0.045) | |
| BMI | | | | | | | | | | | -0.019** | -0.114** |
| | | | | | | | | | | | (0.005) | |
| Alcohol 1 (ref. 0) | | | | | | | | | | | -0.007 | -0.003 |
| | | | | | | | | | | | (0.058) | |
| Alcohol 2–5 (ref. 0) | | | | | | | | | | | 0.047 | 0.022 |
| | | | | | | | | | | | (0.058) | |
| Alcohol 6+ (ref. 0) | | | | | | | | | | | -0.021 | -0.007 |
| | | | | | | | | | | | (0.078) | |
| Smoker | | | | | | | | | | | -0.109* | -0.059* |

(*Continued*)

**Table 4.** (Continued)

| | M1 | | M2 | | M3 | | M4 | | M5 | | M6 | |
|---|---|---|---|---|---|---|---|---|---|---|---|---|
| | Coef. | Beta | Coef. | Beta | Coef. | Beta | Coef. | Beta | Coef. | Beta | Coef. | Beta |
| | | | | | | | | | | | (0.049) | |
| Constant | 3.728** | | 3.419** | | 3.800** | | 3.549** | | 3.613** | | 3.974** | |
| | (-0.074) | | (-0.106) | | (-0.154) | | (-0.149) | | (-0.139) | | (-0.168) | |
| R2 | 0.16 | | 0.20 | | 0.24 | | 0.31 | | 0.40 | | 0.41 | |
| Adj. R2 | 0.15 | | 0.19 | | 0.22 | | 0.30 | | 0.39 | | 0.40 | |
| BIC | 2316.1 | | 2307.2 | | 2296.3 | | 2206.2 | | 2070.4 | | 2086.2 | |

Standardized (beta) and unstandardized coefficients, standard errors, significance tests (t-test: $^*$ $p < 0.05$; $^{**}$ $p < 0.01$), Bayesian information criterion (BIC), the proportion of explained variance ($R^2$), and $R^2$ adjusted for degrees of freedom.

Diagnoses #: The number of conditions respondents was diagnosed with.

Medication #: The number of medication groups respondents was treated with.

Source: QUALITAS 2016/2017 survey (N = 1021).

significantly associated with SRH at the 0.05 significance level. Importantly, these controls explained 16% of the total SRH variance.

Model 2 entered marital status, education, economic activity, and income. Only higher education (tertiary) and higher income were positively linked to better SRH. Considering standardized coefficients, income is a more important predictor of SRH than education but its effect is still moderate. A one-category shift in income produced a 0.04 shift in SRH. All socioeconomic characteristics together contributed only 4% to the explained variance.

Model 3 incorporated biomarkers. In Table 4, we adopted linear measures of biomarkers. In the Appendix, we report supplementary models that used the binary measures, indicating whether the biomarker was over the reference limit and the binary measures adjusted for the medication (Model 1–3 in Table A3 in Appendix). Irrespective of which measurement was used, data indicated that C-reactive protein, glucose, and LDL/HDL ratio were significantly associated with SRH.

Importantly, adding biomarkers to the model with controls and socioeconomic characteristics increased the proportion of explained variance from 20% to 24% (compare Model 2 with Model 3 in Table 4). Furthermore, once biomarkers were included in the model, education ceased to be significant at the 0.05 level. This means that the observed differences among educational groups can be fully attributed to "objective" biomedical measures of health. Model 4 integrated the number of drug classes and the number of medical conditions that the respondents were diagnosed with. Both variables exerted a strong and independent effect on SRH and contributed another 7 percentage points to the explained SRH variance. Among the biomedical measures, the number of drug classes the respondents were treated with constituted the strongest predictor of SRH, followed by the number of conditions (see standardized coefficients).

Model 5 added self-reported mental health status. Standardized coefficients showed that mental health was the strongest predictor of SRH among all variables in the model and was more important than age. Accordingly, this indicator raised the proportion of explained variance by another 9% to 40% of the total explained variance.

Finally, Model 6 entered BMI and behavioral measures. The association between alcohol consumption and SRH was not significant at the 0.05 level, while both BMI and smoking were negatively linked to SRH. However, the standardized coefficients showed that the importance of smoking was relatively weak. The association between BMI and SRH was approximately twice as large as the link between smoking and SRH. Importantly, although BMI and smoking were negatively associated with SRH, they contributed to only 1% of the explained variance.

Overall, the final model, shown in Table 4, showed that mental health and age were the strongest predictors of SRH. These two indicators were followed by the number of drug classes, medical conditions, and BMI. Among the socioeconomic indicators, only income remained statistically significant at the 0.05 level, but it exerted a lower influence than biomedical measures (see standardized coefficients in Model 6 in Table 4). Altogether, Model 6 (Table 4) explained 41% of the SRH variance.

## Discussion and conclusions

This study addressed the question of the concurrent validity of self-rated health in a nationally representative sample from the Czech Republic. Our approach was based on the assumption that an indicator is a valid measure of the outcome if the indicator produces variations in the outcome [41]. In particular, this study addressed the question of the extent to which SRH varies with social conditions compared to health conditions and whether it can be used as an indicator of "true" health in the Czech sample. Thus, we explored how much SRH variance is explained by biomedical, psychological, and social indicators.

The analysis showed that SRH variance can be attributed largely to mental and physical health indicators. This finding suggests that SRH is a valid indicator of "true" health in the Czech sample and that it is a valid and reliable measure of medical burden. At the same time, the results imply that SRH cannot be equated with a narrow biomedical understanding of health. It is rather an indicator of health as a state of complete physical, mental and social well-being. In the bivariate analysis, mental health was by far the most important predictor of SRH (with 14% of variance explained). In the multivariate analysis, biomedical measures (biomarkers, the number of medical conditions, and medication) contributed to around 11% of the variance, while mental health explained around 9% of the variance. Thus, our data suggest that both physical and mental well-being are key dimensions affecting SRH.

The analysis also showed that the social characteristics altogether (marital status, economic activity, education, and income) contributed to only around 5% of explained variance in SRH. However, this does not mean that social inequalities in health are not important in the Czech Republic. Both education and income were significantly linked to SRH and more educated and better off individuals reported better health than those with less education and lower incomes. However, this paper showed that the educational differences ceased to be significant once biomedical and mental health indicators were included in the model. This means that the observed educational differences in SRH reflect differences in "true" health across educational groups. The effect of income persisted after controlling for biomarkers, medication, health conditions, and mental health. Nevertheless, it is important to note that the size of the coefficient for income was reduced significantly once the biomedical and mental status of the individual was considered. The effect was income was particularly visible when the mental health indicator was entered into the model.

This study was not without limitations. First, the data provided only a limited number of biomarkers. Past studies have found that SRH was correlated with markers not available in our dataset, such as hemoglobin and white cell count [15]. Prior research has also reported that heart rate variability was more strongly associated with SRH than inflammatory markers [11]. Such information was also not available in the data. It is possible that more comprehensive measures of respondents' health status would significantly increase the proportion of explained variance.

Second, this study utilized the WHO version of the SRH item ranging from "very good" to "very bad" that is widely used in European surveys. Compared to the US version ranging from "excellent" to "poor", the WHO wording of SRH discriminates better at the positive end but

generally shows less variation and has a less symmetric distribution [59]. Thus, it is possible that our results are affected by the choice of this particular scale. For example, it is possible that the relative role of sociodemographic characteristics would be more pronounced if the more symmetric US version of SRH was used.

Third, it uses cross-sectional data. Thus, our analysis focuses on associations between SRH and other predictors without addressing the issue of causality. There is an ongoing discussion on stability and change in SRH. On the one hand, SRH might be influenced by the individual's transitory standing concerning health status. On the other hand, it might be affected by the enduring self-concept of SRH [60].

Fourth, the data were collected using quota sampling. This method selects individuals with a specific demographic profile that matches the target population (i.e., sex, age, education). It is nonprobability (purposive) sampling in which the interviewers have discretion over who is included. Thus, we need to note that the sample might be biased and it is not possible to estimate the sampling error. For example, it is possible that more health-conscious individuals are over-represented in the sample.

Finally, our paper included only a selected number of indicators that were available in the Qualitas data. Thus, some well-known determinants of SRH (such as functional ability, disability) were not included in our model [61]. It is likely that their inclusion would increase the proportion of explained variance but we cannot determine to what extent. Also, functional limitations are likely to reflect both the physical and mental dimensions of health but also socio-economic conditions of life. Thus, future research should aim to include also functional ability in the model.

## Supporting information

**S1 Appendix. This appendix contains Figs A1–A4 and Tables A1–A4.**
(DOCX)

## Author Contributions

**Conceptualization:** Dana Hamplová.

**Investigation:** Dana Hamplová, Jan Klusáček, Tomáš Mráček.

**Supervision:** Dana Hamplová.

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
