## [Decision Letter · Decision Letter 0]

11 Aug 2021

PONE-D-21-14590

What contributes to the assessment of Self-Rated Health? The relative importance of physiological, mental, and socio-economic factors

PLOS ONE

Dear Dr. Hamplova,

Thank you for submitting your manuscript to PLOS ONE. After careful consideration, we feel that it has merit but does not fully meet PLOS ONE’s publication criteria as it currently stands. Therefore, we invite you to submit a revised version of the manuscript that addresses the points raised during the review process.

Your paper was reviewed by two experts in this area of research.  They have both pointed out concerns with the study sampling, with descriptive and missing data, and with the methods used for analysis.  Notably, they both make the point that the order of entry of the variables into the model is determinative of what the contributions of those variables are to R2.  The issue of data availability is also a concern that you should address as fully as you are able to.

We look forward to receiving your revised manuscript.

Kind regards,

Ellen L. Idler

Academic Editor

PLOS ONE

Journal Requirements:

Acknowledgments: This work was supported by the Czech Academy of Sciences, project “QUALITAS - Wellbeing in health and disease”, which is the part of the Strategy of Czech Academy of Sciences AV21: “Top research in the public interest”,. The authors thank to all the participants who took part in the study and provided blood samples. They are also grateful to employees of SYNLAB for the blood sampling and analyses of biomarkers and to agencies MEDIAN, STEM/MARK and CVVM for questionnaire data collection.

4. Please amend your authorship list in your manuscript file to include the author names.

5. Please amend your list of authors on the manuscript to ensure that each author is linked to an affiliation. Authors’ affiliations should reflect the institution where the work was done (if authors moved subsequently, you can also list the new affiliation stating “current affiliation:….” as necessary).

Reviewers' comments:

Reviewer's Responses to Questions

**Comments to the Author**

1. Is the manuscript technically sound, and do the data support the conclusions?

Reviewer #1: Partly

Reviewer #2: Partly

2. Has the statistical analysis been performed appropriately and rigorously? 

Reviewer #1: No

Reviewer #2: Yes

3. Have the authors made all data underlying the findings in their manuscript fully available?

Reviewer #1: No

Reviewer #2: No

4. Is the manuscript presented in an intelligible fashion and written in standard English?

Reviewer #1: Yes

Reviewer #2: No

5. Review Comments to the Author

Reviewer #1: The aim of this study is to determine the relative importance of physical health, mental health, and SES in SRH in a sample of adult respondents from the Czech Republic.

This submission was easy to read. I am really sorry to say, however, that there are major methodological weaknesses.

The first contribution, the authors state, is to confirm the validity of SRH in this sample. It not entirely clear either what this means (what type of validity?), nor how exactly it was approached methodologically. Moreover, it is my impression that Czech respondents were included in a number of pan-European surveys of population health where SRH was the focus. I am sorry – I don’t know this literature but a quick scholar google search brings up a number of papers. Perhaps they aren’t 100% relevant, but a brief literature review on SRH in Czech respondents would be helpful.

The second aim was to determine the relative importance of the three factors (physical health, mental health, and SES). However, the authors did this by estimating nested OLS models and looked at the increases in R squared. There are a number of issues.

1. Quite concerning: the authors state in the abstract that “Biomarkers (C-reactive protein, blood glucose, triglyceride, low density lipoprotein, high density lipoprotein), self-reported information on medical conditions, and BMI explain 27% of total variance in SRH.” That’s plain wrong, I am sorry. The physical health indicators explain an additional 11% beyond the 16% explained by age and sex.

2. Similarly: the authors conclude that [SES] “contributed to only around 2% of the explained variance.” This is also misleading: it’s 2% after including all measures of physical and mental health measures, which, one might hypothesize, are the mechanisms through which SES and SRH are linked, at least partly.

3. The methods section raised numerous questions.

a. What was the response rate?

b. Were weights used in the analyses?

c. Who is in this sample, and how does the sample differ from the Czech population overall? What is the age range in particular, given that age is, by far, the most ‘important’ covariate of SRH?

d. In Marital status, does “single” mean “never married” or “not married or cohabiting”?

e. How appropriate is OLS for the analysis of SRH? Did the authors conduct any diagnostics?

f. There appears to be a lot of missingness. First, the authors do not discuss it beyond noting 21% missing values on income. What was missingness on other variables? Then they lay out what appears an irregular set of steps to deal with the missingness, culminating with a nonstandard way of apparently hand-averaging 25 imputations and plugging these averages into the models. That does not appear any better than a random realization from multiply imputed datasets and fundamentally fails to account for the increased uncertainty due to the missingness. Perhaps I misunderstand – in that case, could the authors clarify the procedures?

g. For the biomarkers, how were respondents on medication treated? That is, fasting blood glucose can be normal because it’s normal or because an individual is on diabetes medication. Same with triglycerides and statins or other cholesterol medications.

h. It would be good to add an ANOVA test for difference in biomarkers by SRH levels.

4. The mental health indicator is only the CES-D scale for psychological distress (used to measure depressive symptoms). While sleep quality is included, too, this is not a measure of mental health. Or perhaps, in absence of other indicators, it could serve as a proxy for some dimension of mental health but the authors do not explain or motivate this unusual indicator.

5. Why is there no descriptive table, even in the Appendix?

6. Given that the sample appears to be drawn from two locations, why is there no control for location?

7. The Figure 1 is a clear and compelling way to present the results. However, why is education included as a linear covariate? Table 2 suggests its effects are nonlinear. And why is there no indication about how income was included?

8. The Figure might be more informative if, instead of bivariate models, it was based on age-adjusted models (or age and sex).

9. The analysis includes BMI and sleep quality, and classifies them as physical and mental health measures, respectively. There’s no indicator of smoking or alcohol use, which makes me wonder why these behaviors are not included. The authors may push back that they didn’t set out to study health behaviors, which would be fair. However, it might be helpful to discuss this apparent omission at least in the discussion section.

10. Why are SES measures included in the Table 2 last? That would be very important to justify. I would expect to first control for SES (after age and sex) as the fundamental determinant of the other health measures. If done that way, perhaps the increase in proportion explained would still be modest, but it’s important to show what it would be.

Reviewer #2: The current manuscript describes a study based on data from a survey of adults in the Czech Republic. The participants filled in a questionnaire and provided a blood sample. This enabled the researchers to study the predictors of self-rated health (SRH) status and compare the relative contribution of several types of predictors: biomedical, psychological, and social indicators. While the predictors of SRH have been investigated in many studies, only a few included biomarkers. Those mostly used analyses that did not yield an exact estimate of the amount of variance explain by each type of predictor. Thus, analyses based on linear regression models of data that includes biomarkers may provide a worthwhile contribution to this literature. However, there are several issues which the authors could better address.

First and foremost is the determination of the relative contribution of each type of predictors. The authors focus on the addition to the adjust R2 with each additional step. This is greatly affected by the order of entry of the variables into the model. The actual unique contribution of each set of variables in the final model differs from its contribution in the step when it was first entered and is not reported. There was no trial of a different order of entry into the model. Alternatively, better explain why this order of entry of predictors was chosen. Is the aim to determine what part of SRH reflects physiological markers and disease diagnoses and what part is related to psychological measures, social indicators, and demographics? Why was this order of variables chosen? Or is the aim to identify health inequalities – in which case this aim deserves a more detailed explanation. The analyses are interesting, relevant to the question asked and, as the authors wrote, not often reported in this way (in studies including biomarkers), yet could be better explained.

Several additional issues:

The introduction briefly describes existing research on the topic. Despite the lack of a clear comparison between types of predictors in their contribution to SRH, there are several additional articles that included a range of predictors and could be cited, to provide a fuller picture of the existing literature. For example, Goldman et a. 2009 (doi:10.1016/S1047-2797(03)00077-2), Anreasson et al., 2013 (DOI: 10.1177/1359105311435428).

The study procedure is barely described – much information is missing on the way potential participants were identified, approached and recruited to the study, where they provided the data, and on response rates.

CES-D – the questionnaire originally included 20 items. Shorter 10- or 11-item versions are often used and have been validated. A four item version is not commonly used and it would be appropriated to add information supporting its validity.

Table 2 – if the aim is to compare the contribution of different predictors, it makes more sense to report standardized regression coefficients.

Study limitations – the study is cross-sectional. Longitudinal data may reveal additional information; some well-known correlates of SRH were not included, for example, physical functioning, cognitive functioning, positive emotions.

Corrections

Please correct typos and language errors on lines 45, 58, 93, 137, 156 (+define the abbreviation for CRP), 160, 170, 171, 181, 235, 263, 282, 296, number of diagnoses (in Table 2).

Correct author names in reference 9 (Leshem-Robinow et al.).

6. PLOS authors have the option to publish the peer review history of their article (what does this mean?). If published, this will include your full peer review and any attached files.

Reviewer #1: No

Reviewer #2: No

---

## [Author Response · Author response to Decision Letter 0]

30 Sep 2021

CHANGES IN RESPONSE TO REVIEWS

First of all, we would like to thanks all reviewers for their time spent on reviewing our manuscript and for the valuable and thoughtful comments towards improving our manuscript. Before we address these concerns, point by point, we would like to describe the main conceptual changes.

Both reviewers expressed concerns about the order, in which variables entered the analysis. They argued that the stepwise building of the model and the order of variables block might significantly affect the results. To address these concerns, we changed our analytical strategy. First, we analyze the blocks of variables separately only controlling for age, sex, and location. Second, we built the final model using these blocks. As suggested by Reviewer 1, we start with the sociodemographic variables, and biomedical measures are entered only afterward.

REVIEWER #1: 

The aim of this study is to determine the relative importance of physical health, mental health, and SES in SRH in a sample of adult respondents from the Czech Republic.

This submission was easy to read. I am really sorry to say, however, that there are major methodological weaknesses. The first contribution, the authors state, is to confirm the validity of SRH in this sample. It not entirely clear either what this means (what type of validity?), nor how exactly it was approached methodologically. 

We specify that we measure concurrent validity of SRH, i.e. the extent to which this indicator correlates with established measures of health. We also specify the measure that we are interested in in the relative contribution of different types of predictors.

Moreover, it is my impression that Czech respondents were included in a number of pan-European surveys of population health where SRH was the focus. I am sorry – I don’t know this literature but a quick scholar google search brings up a number of papers. Perhaps they aren’t 100% relevant, but a brief literature review on SRH in Czech respondents would be helpful.

We added a brief literature review on the Czech Republic. There is a number of studies using SRH but the research on validity of SRH is limited.

The second aim was to determine the relative importance of the three factors (physical health, mental health, and SES). However, the authors did this by estimating nested OLS models and looked at the increases in R squared. There are a number of issues.

1. Quite concerning: the authors state in the abstract that “Biomarkers (C-reactive protein, blood glucose, triglyceride, low density lipoprotein, high density lipoprotein), self-reported information on medical conditions, and BMI explain 27% of total variance in SRH.” That’s plain wrong, I am sorry. The physical health indicators explain an additional 11% beyond the 16% explained by age and sex.

This section was completely changed based on the other comments from both reviewers. The abstract has been re-written.

2. Similarly: the authors conclude that [SES] “contributed to only around 2% of the explained variance.” This is also misleading: it’s 2% after including all measures of physical and mental health measures, which, one might hypothesize, are the mechanisms through which SES and SRH are linked, at least partly.

Again, the section has been re-written and formulation changed.

3. The methods section raised numerous questions.

a. What was the response rate? b. Were weights used in the analyses? c. Who is in this sample, and how does the sample differ from the Czech population overall? What is the age range in particular, given that age is, by far, the most ‘important’ covariate of SRH?

We added a new section with the description of the dataset. As we use quota sampling, it is not possible to determine response rates or use weights.

d. In Marital status, does “single” mean “never married” or “not married or cohabiting”?

We specify that “single” means “currently living without a partner”.

e. How appropriate is OLS for the analysis of SRH? Did the authors conduct any diagnostics?

We report diagnostics of the OLS in the Appendix. We also refer to existing studies using OLS to study SRH.

f. There appears to be a lot of missingness. First, the authors do not discuss it beyond noting 21% missing values on income. What was missingness on other variables? Then they lay out what appears an irregular set of steps to deal with the missingness, culminating with a nonstandard way of apparently hand-averaging 25 imputations and plugging these averages into the models. That does not appear any better than a random realization from multiply imputed datasets and fundamentally fails to account for the increased uncertainty due to the missingness. Perhaps I misunderstand – in that case, could the authors clarify the procedures?

In general, income was the only variable with a larger proportion of missing data. We acknowledge that using a simple mean of imputed values is problematic. Thus, we employ three methods to cope with the missing data. First, we estimate the results only for cases without missing information. Second, we employ multiple imputations and we use the mean value of the imputed income (as it was done in the original version of the paper). Third, we estimate the set of regression models with the imputed data and we report on the distribution of R2 values.

g. For the biomarkers, how were respondents on medication treated? That is, fasting blood glucose can be normal because it’s normal or because an individual is on diabetes medication. Same with triglycerides and statins or other cholesterol medications.

In the revised version of the paper, we included a new variable referring to medications. As for the biomarkers, we distinguished those below and over the reference limit. In the analysis, we used both crude and adjusted measures. The crude measure refers to the number of biomarkers over the reference limits. The adjusted measure considers whether the respondent takes medication for the given condition. 

h. It would be good to add an ANOVA test for difference in biomarkers by SRH levels.

We added ANOVA test for differences in biomarkers in Table 2.

4. The mental health indicator is only the CES-D scale for psychological distress (used to measure depressive symptoms). While sleep quality is included, too, this is not a measure of mental health. Or perhaps, in absence of other indicators, it could serve as a proxy for some dimension of mental health but the authors do not explain or motivate this unusual indicator.

We adjusted the analysis. The original CES-D scale contains an item on sleep disturbances, but this item has not been included in the Qualitas questionnaire. Instead, it was replaced by a different measure of sleep quality. In the revised version of the paper, we run a measurement model (confirmatory factor analysis using SEM technique) to test whether our measure of sleep quality could be included in the measure of mental health. Indeed, the measurement model confirmed that all 4 items from CES-D and the sleep quality item load on one latent variable (mental health – the full measurement model is reported in the Appendix). Thus, in the revised version of the paper, we use one indicator of mental health including all 5 items.

5. Why is there no descriptive table, even in the Appendix?

We added a descriptive table (Table 1).

6. Given that the sample appears to be drawn from two locations, why is there no control for location?

All models now control for the location.

7. The Figure 1 is a clear and compelling way to present the results. However, why is education included as a linear covariate? Table 2 suggests its effects are nonlinear. And why is there no indication about how income was included?

Education is now included as a categorical variable. In the figure, we denote which variable is linear, binary, and categorical.

8. The Figure might be more informative if, instead of bivariate models, it was based on age-adjusted models (or age and sex).

We adjusted the results for age, sex, and location.

9. The analysis includes BMI and sleep quality, and classifies them as physical and mental health measures, respectively. There’s no indicator of smoking or alcohol use, which makes me wonder why these behaviors are not included. The authors may push back that they didn’t set out to study health behaviors, which would be fair. However, it might be helpful to discuss this apparent omission at least in the discussion section.

Alcohol intake and smoking are now incorporated into the analysis.

10. Why are SES measures included in the Table 2 last? That would be very important to justify. I would expect to first control for SES (after age and sex) as the fundamental determinant of the other health measures. If done that way, perhaps the increase in proportion explained would still be modest, but it’s important to show what it would be.

In the revised version of the paper, we use a different analytical strategy. We first enter the blocks of variables independently. In the final model, we started with the controls, SES, and then add other health measures.

REVIEWER #2: 

The current manuscript describes a study based on data from a survey of adults in the Czech Republic. The participants filled in a questionnaire and provided a blood sample. This enabled the researchers to study the predictors of self-rated health (SRH) status and compare the relative contribution of several types of predictors: biomedical, psychological, and social indicators. While the predictors of SRH have been investigated in many studies, only a few included biomarkers. Those mostly used analyses that did not yield an exact estimate of the amount of variance explain by each type of predictor. Thus, analyses based on linear regression models of data that includes biomarkers may provide a worthwhile contribution to this literature. However, there are several issues which the authors could better address. 

First and foremost is the determination of the relative contribution of each type of predictors. The authors focus on the addition to the adjust R2 with each additional step. This is greatly affected by the order of entry of the variables into the model. The actual unique contribution of each set of variables in the final model differs from its contribution in the step when it was first entered and is not reported. There was no trial of a different order of entry into the model. Alternatively, better explain why this order of entry of predictors was chosen. Is the aim to determine what part of SRH reflects physiological markers and disease diagnoses and what part is related to psychological measures, social indicators, and demographics? Why was this order of variables chosen? Or is the aim to identify health inequalities – in which case this aim deserves a more detailed explanation. The analyses are interesting, relevant to the question asked and, as the authors wrote, not often reported in this way (in studies including biomarkers), yet could be better explained.

In the revised version of the paper, we adopted a new analytical strategy that uses a different order of entering variables into the model (see the description above)

Several additional issues:

The introduction briefly describes existing research on the topic. Despite the lack of a clear comparison between types of predictors in their contribution to SRH, there are several additional articles that included a range of predictors and could be cited, to provide a fuller picture of the existing literature. For example, Goldman et a. 2009 (doi:10.1016/S1047-2797(03)00077-2), Anreasson et al., 2013 (DOI: 10.1177/1359105311435428).

The articles are added to the literature review.

The study procedure is barely described – much information is missing on the way potential participants were identified, approached and recruited to the study, where they provided the data, and on response rates.

We added better description of survey and participants’ characteristics.

CES-D – the questionnaire originally included 20 items. Shorter 10- or 11-item versions are often used and have been validated. A four item version is not commonly used and it would be appropriated to add information supporting its validity.

In the Appendix, we report a measurement model (SEM) and we show that the 4-item scale has a very good internal consistency. We comment on this fact in the main body of the paper.

Table 2 – if the aim is to compare the contribution of different predictors, it makes more sense to report standardized regression coefficients.

Standardized coefficients are included in all tables.

Study limitations – the study is cross-sectional. Longitudinal data may reveal additional information; some well-known correlates of SRH were not included, for example, physical functioning, cognitive functioning, positive emotions.

We acknowledged this limitation in the study limitations.

Corrections

Please correct typos and language errors on lines 45, 58, 93, 137, 156 (+define the abbreviation for CRP), 160, 170, 171, 181, 235, 263, 282, 296, number of diagnoses (in Table 2).

The article has been edited by a professional service specializing in academic writing, all abbreviations are defined. 

Correct author names in reference 9 (Leshem-Robinow et al.).

The name as it is printed on the publication is used.

---

## [Decision Letter · Decision Letter 1]

15 Nov 2021

PONE-D-21-14590R1Assessment of self-rated health: The relative importance of physiological, mental, and socioeconomic factorsPLOS ONE

Dear Dr. Hamplova,

Thank you for submitting your manuscript to PLOS ONE. After careful consideration, we feel that it has merit but does not fully meet PLOS ONE’s publication criteria as it currently stands. Therefore, we invite you to submit a revised version of the manuscript that addresses the points raised during the review process. Here are the reviews from the second round of reviews, following your major revision of the paper.  One of the original reviewers was no longer available and I invited a new reviewer.  The first reviewer, who has also reviewed the revision, has positive feedback on the revisions, but also some remaining issues with the overall message and meaning of the paper.  Those concerns are very much in line with the review by the second, new reviewer.  Both reviewers point out that, while the analysis has been carried out mostly well, there is little discussion of the meaning of the findings, and insufficient attempt to place the findings in the larger research literature on the topic. Given that you have made strong improvements to the first submission, I would encourage you to turn to the detailed comments of these two highly expert reviewers, to make additional improvements in the manuscript.

We look forward to receiving your revised manuscript.

Kind regards,

Ellen L. Idler

Academic Editor

PLOS ONE

Reviewers' comments:

Reviewer's Responses to Questions

**Comments to the Author**

1. If the authors have adequately addressed your comments raised in a previous round of review and you feel that this manuscript is now acceptable for publication, you may indicate that here to bypass the “Comments to the Author” section, enter your conflict of interest statement in the “Confidential to Editor” section, and submit your "Accept" recommendation.

Reviewer #1: (No Response)

Reviewer #3: (No Response)

2. Is the manuscript technically sound, and do the data support the conclusions?

Reviewer #1: Yes

Reviewer #3: Partly

3. Has the statistical analysis been performed appropriately and rigorously? 

Reviewer #1: Yes

Reviewer #3: Yes

4. Have the authors made all data underlying the findings in their manuscript fully available?

Reviewer #1: No

Reviewer #3: No

5. Is the manuscript presented in an intelligible fashion and written in standard English?

Reviewer #1: Yes

Reviewer #3: No

6. Review Comments to the Author

Reviewer #1: The authors were highly responsive to all comments and the extensive revision yielded a much stronger paper. I am sorry, however, to pose several residual questions.

Introduction last paragraph: what is “this social context” in line 83? And in lines 88-90, do the authors refer to Czech republic?

Methods: I didn’t understand why the quota sampling invalidates the question about response rate. Presumably some individuals who were approach did not participate: what was the response rate?

Approach: about the multiple imputation, page 6 lines 131-2. How or where is the distribution of the 25 R squareds reported?

Minor: I would suggest referring to the two locations as “location,” not “region.”

Results: the fact that only 2 out of >1,000 people assessed their health as “very bad” suggests that the SRH item, or the labels used for its 5 categories in the Czech language, capture something different than the English labels of excellent, very good, good, fair, or poor. This should be discussed as a limitation.

Results: more importantly, the results are now much expanded but a bit difficult to follow. It seems the authors incorporated a lot of different approaches and robustness checks. I think the results section would be much stronger if they only retained one approach/specification and told the story with it; all the robustness checks could be in the appendix.

There is little discussion in general. One issue that I was hoping to see was to engage with the issue that socioeconomic factors are more distal characteristics, that may influence SRH via the more proximate factors, such as health conditions or biological risk variables.

More broadly, there is little discussion that would help readers understand the results. What do the findings mean for SRH, or for the Czech population health, or for population-health research? Why might it be that this study found such a weak relationship between SES and SRH? Etc.

Reviewer #3: The study examines associations of physiological, mental and socioeconomic factors with self-rated health, specifically “the relative contribution of biomedical and social characteristics to an individual´s SRH assessment” in a population sample from the Czech republic. The topic is basically relevant. Yet the paper has a major flaw in that the basic idea of the study seems to be missing. It remains unclear why these associations are important to know, why they are analyzed here, and what should we learn from the results. It is striking that the crucial part in all studies, interpretation of findings, is largely missing. The Discussion only repeats the statistical results and discusses limitations, concerning mainly data.

There are several starts in the Introduction that let the reader believe that this is may be the theme of the study, the perspective from which the associations/ relative explanatory powers are studied, but none of these themes seems to have guided the analyses, nor are they discussed in Discussion. The validity of SRH is mentioned in several places in the paper. Yet is remains unclear, what validity the authors have in their minds; the validity of SRH as a measure of objective health?; its validity as predictor of death?, or something else. On p 88-90 validity against biomedical data is mentioned. Also, there are references in Introduction to the extent to which SRH may rely on social experience, to what extent it is a measure of social determination of health, and also a reference to health inequalities. These perspectives would, if followed more, lead to different specific research question and in all likelihood also to different analytic designs. For instance, if SRH were considered as a measure of health inequalities, it would be important to analyze, or at least discuss, whether it reflects different objective health (disease, functioning…) between different (socioeconomic?) groups, or whether and to what extent it reflects difference in the way different groups evaluate their health. The conclusion in the Abstract concludes that SRH likely is a valid measure of physiological and mental health in the Czech sample and the observed differences were likely to reflect inequalities in bodily and mental functions between social groups. But in the analyses, there is no comparison whatsoever between social groups. If this were the topic of the study, one proper design for the analysis would be to see whether the differences in SRH between different social groups follow the differences in physical and mental health in these groups.

The Introduction says that the aim of the analyses is to determine how much variance of SRH can be explained by biomedical (sometimes called physiological) , psychological, and social indicators. This, again, is a different perspective from the perspectives of validity and inequality, but this perspective either is clearly justified; the reader expects the authors to explain why we need to know this. Another big question is, what is the authors´ understanding on the possible mechanisms on the associations of the selected factor groups with SRH. It is justified to say that the mechanisms on why diseases and depression explain variance of SRH are different from the mechanisms on why income explains it.

The problems described above lead me to suggest that the authors would plan the thematization, research questions, and analytic design again from the beginning, in order to use the data available in a way that could give a real contribution to the present literature.

If the authors decide to continue their work on this paper and maintain the focus on the explanatory power on different variable groups, they may also consider the following comments:

• It is problematic to claim (p3) that ”what contributes to an individual´s assessment of SRH remains largely unknown”. Today there are several studies combining empiric data with conceptual understanding and creating useful frameworks on what and how contributes to self-assessments of health. It is true that there is no exhaustive and comprehensive list of contributing factors, but then, there cannot be, this is an inherent and necessary characteristic of the self-rated health which is a subjective construction of everything that a person considers as belonging to his/her “health”.

• As the authors speak about the relative contribution of biomedical and social characteristics, I am sure many readers expect to learn about the relative contribution of these variable groups. Yet the biomedical and social variables are not analyzed as groups but only as individual variables. As such, the empirical findings are not novel. The authors write (“p4 ..”no study has tested the concurrent validity of SRH, ie the extent to which this indicator correlates with established measures of health, using biomedical data”. This is clearly wrong. There are numerous studies that analyze the associations of diseases, functioning, symptoms, medication with SRH, and recently also several studies that analyze the association of SRH with indicators measured from blood. Also the associations with a variety of different social variables, including socio-economic, and behavioral variables with SRH have been studied. Mainly the findings of this manuscript are in line with earlier studies in that highest associations are found for diseases, other health variables, and also socioeconomic factors.

• The sequence of analyses is hard to follow. As it is not groups of variables but individual variables belonging to different groups that are studied, why is it necessary to show the basic associations in separate tables? Why not first show the individual associations in one table and then try to analyze the contributions of different groups of variables? As to the individual explanatory factors, it would be fair to take into account and at least discuss the fact that the associations and explanatory powers depend not only to the content of the indicator but also on the number and categories of the variables included in the indicator.

• To an international audience it is not a very good justification of a study that these associations have not earlier been studied in a Czech population. Numerous studies show that SRH as a variable behaves largely in the same way in different countries, different population groups etc. If there is no good reason to believe that in Czech the association would be different, the justification of the study should be based on its scientific novelty and message. Of course, Czech population can work as well as any other for this purpose.

.

7. PLOS authors have the option to publish the peer review history of their article (what does this mean?). If published, this will include your full peer review and any attached files.

Reviewer #1: No

Reviewer #3: No

---

## [Author Response · Author response to Decision Letter 1]

2 Feb 2022

We would like to thank both reviewers for their time and helpful comments that helped to improve the manuscript. We hope that we address all points raised in the reviews.

Before we address each point in detail, we would like to describe the major changes to the paper. First, we revised the introductory part to make it more focused and to explain the research agenda more clearly. Second, we followed the advice of Reviewer 1 who suggested to move much of the results to the Appendix. Thus, the result section is more concise and we believe it is easier to follow. Third, we expanded the discussion and we point out the major implications of our findings.

Reviewer #1: 

The authors were highly responsive to all comments and the extensive revision yielded a much stronger paper. I am sorry, however, to pose several residual questions.

Introduction last paragraph: what is “this social context” in line 83? And in lines 88-90, do the authors refer to Czech republic?

The sentence has been reformulated to make it clear that we talk about the Czech Republic.

Methods: I didn’t understand why the quota sampling invalidates the question about response rate. Presumably some individuals who were approach did not participate: what was the response rate?

The information on how many respondents from the originally selected sample is provided.

Approach: about the multiple imputation, page 6 lines 131-2. How or where is the distribution of the 25 R squareds reported?

The section is moved to the Appendix.

Minor: I would suggest referring to the two locations as “location,” not “region.”

The term region has been replaced by “location” in the whole text.

Results: the fact that only 2 out of >1,000 people assessed their health as “very bad” suggests that the SRH item, or the labels used for its 5 categories in the Czech language, capture something different than the English labels of excellent, very good, good, fair, or poor. This should be discussed as a limitation.

In the discussion, we point out that this study utilized the WHO version of the SRH item ranging from “very good” to “very bad” that is widely used in European surveys. This item has generally better discrimination power at the positive end. 

Results: more importantly, the results are now much expanded but a bit difficult to follow. It seems the authors incorporated a lot of different approaches and robustness checks. I think the results section would be much stronger if they only retained one approach/specification and told the story with it; all the robustness checks could be in the appendix.

As suggested, we moved a significant part of the result section to the Appendix and we believe that the flow of the text is easier to follow in this version of the paper.

There is little discussion in general. One issue that I was hoping to see was to engage with the issue that socioeconomic factors are more distal characteristics, that may influence SRH via the more proximate factors, such as health conditions or biological risk variables.

More broadly, there is little discussion that would help readers understand the results. What do the findings mean for SRH, or for the Czech population health, or for population-health research? Why might it be that this study found such a weak relationship between SES and SRH? Etc.

We expanded the discussion in the suggested direction.

Reviewer #3: 

The study examines associations of physiological, mental and socioeconomic factors with self-rated health, specifically “the relative contribution of biomedical and social characteristics to an individual´s SRH assessment” in a population sample from the Czech republic. The topic is basically relevant. Yet the paper has a major flaw in that the basic idea of the study seems to be missing. It remains unclear why these associations are important to know, why they are analyzed here, and what should we learn from the results.

We revised the introductory part of the paper and we hope that the basic idea of the study is expressed better. We try to explain as to why it is relevant to validate the SRH item in different contexts.

It is striking that the crucial part in all studies, interpretation of findings, is largely missing. The Discussion only repeats the statistical results and discusses limitations, concerning mainly data.

The results section is completely revised. Much of the detailed findings have been moved to the appendix. We also completely revised the discussion to make explain the contribution and implications of the study.

There are several starts in the Introduction that let the reader believe that this is may be the theme of the study, the perspective from which the associations/ relative explanatory powers are studied, but none of these themes seems to have guided the analyses, nor are they discussed in Discussion. The validity of SRH is mentioned in several places in the paper. Yet is remains unclear, what validity the authors have in their minds; the validity of SRH as a measure of objective health?; its validity as predictor of death?, or something else. …..

We completely revised the introductory section and we hope that the aim of the manuscript and the issue of validity is clearer.

If the authors decide to continue their work on this paper and maintain the focus on the explanatory power on different variable groups, they may also consider the following comments:

• It is problematic to claim (p3) that ”what contributes to an individual´s assessment of SRH remains largely unknown”. …

The section is revised.

• As the authors speak about the relative contribution of biomedical and social characteristics, I am sure many readers expect to learn about the relative contribution of these variable groups. Yet the biomedical and social variables are not analyzed as groups but only as individual variables. 

We explain our approach in the method section. Our work is inspired by Hiyoshi et al.’s approach to estimating the joint contribution of several variables. Thus, we do not focus on individual coefficients but the joint contribution to the explained variance.

As such, the empirical findings are not novel. The authors write (“p4 ..”no study has tested the concurrent validity of SRH, ie the extent to which this indicator correlates with established measures of health, using biomedical data”. This is clearly wrong. ….

Here, we refer to the national context. Hopefully, it is clearer in the revised version.

• To an international audience it is not a very good justification of a study that these associations have not earlier been studied in a Czech population. Numerous studies show that SRH as a variable behaves largely in the same way in different countries, different population groups etc. If there is no good reason to believe that in Czech the association would be different, the justification of the study should be based on its scientific novelty and message. Of course, Czech population can work as well as any other for this purpose.

We hope that we explain the contribution of the manuscript clearer.

---

## [Decision Letter · Decision Letter 2]

25 Feb 2022

PONE-D-21-14590R2Assessment of self-rated health: The relative importance of physiological, mental, and socioeconomic factorsPLOS ONE

Dear Dr. Hamplova,

Thank you for submitting your manuscript to PLOS ONE. After careful consideration, we feel that it has merit but does not fully meet PLOS ONE’s publication criteria as it currently stands. Therefore, we invite you to submit a revised version of the manuscript that addresses the points raised during the review process.

Overall, both reviewers were very pleased with the revisions and indicate that it is a much stronger paper.  Reviewer 3 has just a small number of (relatively minor) comments.  If you could attend to them I would plan to send the paper back to only this reviewer, which should speed the process.  Regarding Reviewer 1's reply of "no" to question 4 concerning data availability -- it would seem to be an error since your data are completely available.  I will make certain that this is addressed if it raises issues with the journal.

We look forward to receiving your revised manuscript.

Kind regards,

Ellen L. Idler

Academic Editor

PLOS ONE

Journal Requirements:

Reviewers' comments:

Reviewer's Responses to Questions

**Comments to the Author**

1. If the authors have adequately addressed your comments raised in a previous round of review and you feel that this manuscript is now acceptable for publication, you may indicate that here to bypass the “Comments to the Author” section, enter your conflict of interest statement in the “Confidential to Editor” section, and submit your "Accept" recommendation.

Reviewer #1: All comments have been addressed

Reviewer #3: (No Response)

2. Is the manuscript technically sound, and do the data support the conclusions?

Reviewer #1: Yes

Reviewer #3: Yes

3. Has the statistical analysis been performed appropriately and rigorously? 

Reviewer #1: Yes

Reviewer #3: Yes

4. Have the authors made all data underlying the findings in their manuscript fully available?

Reviewer #1: No

Reviewer #3: Yes

5. Is the manuscript presented in an intelligible fashion and written in standard English?

Reviewer #1: Yes

Reviewer #3: Yes

6. Review Comments to the Author

Reviewer #1: Another thorough and skillful revision, thanks. I have no further comments; the manuscript is a compelling analysis.

Reviewer #3: The manuscript has clearly improved. I only have three minor comments.

P 4 says that face-to-face interviews were conducted, later a questionnaire is mentioned several times. Please clarify.

Due to sampling method, response rate could not be calculated. In all likelihood the sample is not representative for the population in the study areas. This does not hamper the results but would be worth mentioning in Discussion.

A major well-known factor associated with SRH was missing, functioning, functional ability, disability... It would be good to mention this in the Discussion.

7. PLOS authors have the option to publish the peer review history of their article (what does this mean?). If published, this will include your full peer review and any attached files.

Reviewer #1: No

Reviewer #3: **Yes: **Marja Jylhä

---

## [Author Response · Author response to Decision Letter 2]

31 Mar 2022

** P4 says that face-to-face interviews were conducted, later a questionnaire is mentioned several times. Please clarify.

We clarify that a questionnaire was administered by face-to-face mode of data collection.

Due to sampling method, response rate could not be calculated. 

The comment on response rate was included as a reaction to a reviewer comment. It is deleted from the revised manuscript.

** In all likelihood the sample is not representative for the population in the study areas. This does not hamper the results but would be worth mentioning in Discussion.

It is mentioned in the discussion.

** The limitation - A major well-known factor associated with SRH was missing, functioning, functional ability, disability... It would be good to mention this in the Discussion.

It is mentioned in the discussion among the study limitations.

---

## [Editor Report · Decision Letter 3]

4 Apr 2022

Assessment of self-rated health: The relative importance of physiological, mental, and socioeconomic factors

PONE-D-21-14590R3

Dear Dr. Hamplova,

We’re pleased to inform you that your manuscript has been judged scientifically suitable for publication and will be formally accepted for publication once it meets all outstanding technical requirements.

Kind regards,

Ellen L. Idler

Academic Editor

PLOS ONE
---

## [Editor Report · Acceptance letter]

8 Apr 2022

PONE-D-21-14590R3 

Assessment of self-rated health: The relative importance of physiological, mental, and socioeconomic factors 

Dear Dr. Hamplová:

I'm pleased to inform you that your manuscript has been deemed suitable for publication in PLOS ONE. Congratulations! Your manuscript is now with our production department. 

Kind regards, 

on behalf of

Professor Ellen L. Idler 

Academic Editor

PLOS ONE